# A general method for controlling and resolving rotational orientation of molecules in molecule-surface collisions

Oded Godsi[1], Gefen Corem[1], Yosef Alkoby[1], Joshua T. Cantin[2], Roman V. Krems[2], Mark F. Somers[3], Jörg Meyer[3], Geert-Jan Kroes[3], Tsofar Maniv[1] & Gil Alexandrowicz[1]

The outcome of molecule–surface collisions can be modified by pre-aligning the molecule; however, experiments accomplishing this are rare because of the difficulty of preparing molecules in aligned quantum states. Here we present a general solution to this problem based on magnetic manipulation of the rotational magnetic moment of the incident molecule. We apply the technique to the scattering of $H_2$ from flat and stepped copper surfaces. We demonstrate control of the molecule's initial quantum state, allowing a direct comparison of differences in the stereodynamic scattering from the two surfaces. Our results show that a stepped surface exhibits a much larger dependence of the corrugation of the interaction on the alignment of the molecule than the low-index surface. We also demonstrate an extension of the technique that transforms the set-up into an interferometer, which is sensitive to molecular quantum states both before and after the scattering event.

[1] Schulich Faculty of Chemistry, Technion—Israel Institute of Technology, Technion City, Haifa 32000, Israel. [2] Department of Chemistry, University of British Columbia, Vancouver, British Columbia, Canada V6T 1Z1. [3] Leiden Institute of Chemistry, Gorlaeus Laboratories, Leiden University, PO Box 9502, 2300 RA Leiden, The Netherlands. Correspondence and requests for materials should be addressed to G.A. (email: ga232@tx.technion.ac.il).

Molecules that collide with a surface can stick, dissociate or scatter depending on the exact initial state of the molecule and the nature of the molecule–surface interaction. The balance between these different possible outcomes is a key question in many research fields and applications, ranging from industrial heterogeneous catalysis to astrochemistry. A principal contribution towards understanding gas–surface interactions comes from quantum state-resolved experiments that probe the interaction of a molecule with the surface without averaging over many initial and final conditions[1–3]. A particularly important quantum state to resolve is the rotation projection state, which determines the alignment and orientation of the rotational motion of the molecule relative to the surface. It is known that gas–surface interactions can be highly dependent on this stereodynamic feature[1,3–7], and there are various unconfirmed theoretical predictions regarding this dependence[1,8–11]. Whereas the incentive to do so is high, the ability to experimentally resolve the role of rotation projection states in molecule–surface scattering has so far been restricted to a rather small subset of systems for which specific innovative experimental approaches could be applied. These approaches include deflection of paramagnetic and polar molecules, photo-excitation of desorbed molecules and exploiting correlations between velocity and rotational alignment in supersonic beams[1,5,12]. Enabling wider access to experimental benchmarks in this field is an essential step in advancing our understanding of molecule–surface interactions.

Here we present a general approach to tackle this problem, and demonstrate its effectiveness by both probing and controlling the initial rotation projection quantum state of a hydrogen molecule, and correspondingly the outcome of its collision with the surface. Our approach exploits the magnetic moment associated with the rotation of a molecule, a subtle property of each molecule, which has been studied in a series of seminal experiments by Rabi, Ramsey and co-workers[13]. In the absence of unpaired electronic spins, the total wave function of a molecule in a magnetic field is described by a combination of rotation and nuclear spin states[13], which we refer to as the Ramsey states of the molecule. Each of these states responds differently to magnetic fields, and it is this response that allows us to probe and control the different states involved in the scattering event.

## Results

**Experimental.** Our experimental apparatus is based on a helium–spin–echo spectrometer[14], to which modifications were made to transform it into a molecular interference set-up. Similarly to the case of atom (helium-3) interferometry[15], different types of measurements can be performed with this scheme, depending on the choice of magnetic field strength and direction. Here we present two types of molecular beam experiments that we refer to as flux detection and full-interferometry experiments. We start by describing the propagation of the molecules through the first arm of the instrument, which is common to both types of experiments.

As a first application of this approach, we performed experiments with a molecular beam of $H_2$ scattered from a flat (111) and a stepped (115) copper surface. After the supersonic expansion at the source, the beam contains predominantly the two lowest rotational states, para-$H_2$ ($J=0$) and ortho-$H_2$ ($J=1$). The para-$H_2$ component is hereafter disregarded as it is unaffected by magnetic field manipulations and is simply seen as a constant background in our experiments. Ortho-$H_2$ ($J=1$), on the other hand, decomposes into nine Ramsey states, which are the different combinations of the three rotation projection states $m_J = -1,0,1$ and the three nuclear spin states $m_I = -1,0,1$.

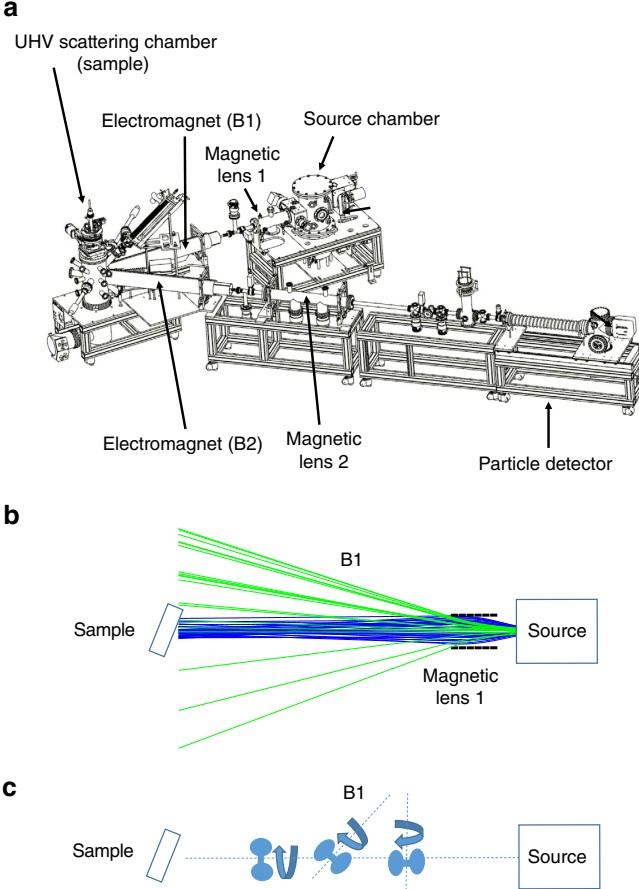

**Figure 1 | Experimental set-up.** (**a**) Dimensional drawing of the apparatus. (**b**) Illustration of the beam deflecting magnetic manipulation discussed in the text. The initial quantum state of the beam is defined by the magnetic lens (lens 1), which focuses and defocuses different quantum states. (**c**) Illustration of the manipulation of the rotation projection states within the B1 field. The effect of the field can be visualized as a precession of the axis of rotation of a molecule.

Figure 1a shows a to-scale drawing of the apparatus with its V-shaped beam line. The first magnetic field the molecular beam enters is a variation on the state-selecting Stern–Gerlach magnet[16] (labelled magnetic lens 1 in Fig. 1a). It is a hexapole magnetic lens[17] within which different quantum states experience different (quantized) forces depending on their total magnetic moment[13] (see ref. 18 for a contemporary interpretation of beam deflection experiments). The schematic drawing in Fig. 1b illustrates this type of magnetic manipulation for the case of ortho-$H_2$ ($J=1$), where the blue lines show molecular beam trajectories for $m_I=1, m_J=1$, which are focused by the magnetic lens. In contrast, the green lines show the hypothetical trajectories of a defocused state $m_I=-1, m_J=-1$, where in practice the deflected molecules hit the vacuum chamber wall and are pumped away by the vacuum system. As we demonstrated in the past for the case of ortho-$H_2O$ molecules[19], a hexapole magnetic lens can be used to focus some molecular quantum states and extract others from the beam line. However, it is important to note that, generally speaking, a magnetic focusing element cannot be used on its own as a method of selecting a pure rotational projection state because of the spin–rotation coupling terms[20] that lead to mixing of the rotational states even in the absence of magnetic fields[21]. Instead, the lens selects

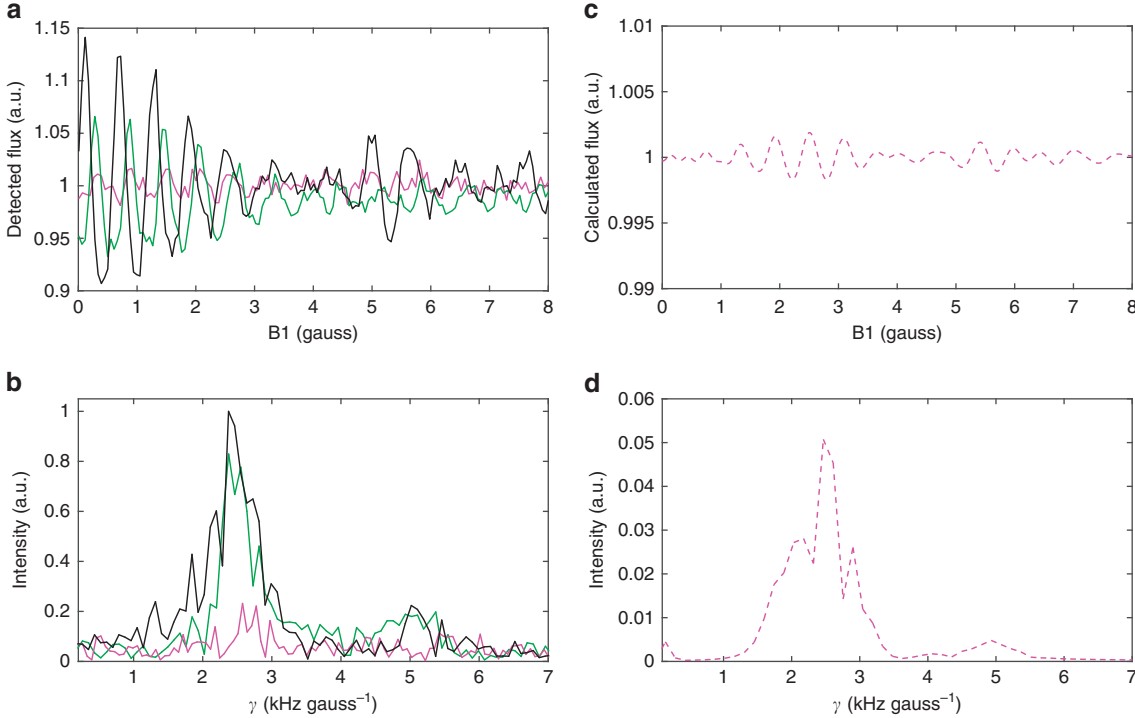

**Figure 2 | Flux detection and oscillation.** (**a**,**b**) Flux detection measurements and corresponding spectra for $H_2$ scattering from copper surfaces. The magenta curves show results for Cu(111), whereas the black and green curves are for Cu(115) at two different crystal azimuths (parallel/perpendicular to the atomic step direction, respectively). As we modify the field B1, we change the quantum state of the impinging molecule and the stereo-selective scattering process leads to clearly resolved flux oscillations. (**c**) Calculated flux oscillations for $H_2$ scattering from Cu(111) using the scattering probabilities mentioned in the text. (**d**) Spectrum of flux oscillation calculation shown in **c**. The horizontal axis of the spectra shown in **b** and **d**, was plotted using frequency per magnetic field units, labelled as $\gamma$.

the initial quantum state of the molecule, which then evolves into a superposition of $m_I, m_J$ states as the molecule travels through the beam line, as discussed in more detail in Supplementary Note 3.

The second crucial element is an electromagnet (B1), which is used to control the evolution of the quantum (superposition) state mentioned above, according to the Hamiltonian $\hat{H} = \frac{\hbar^2 k^2}{2m_{H_2}} + \hat{H}_R(B1)$. Here $\hat{H}_R(B1)$ is the Ramsey Hamiltonian defined in ref. 20, $\hbar$ is the reduced Planck constant, $m_{H_2}$ is the $H_2$ mass and B1 is the magnetic field. During the propagation of the beam within B1, the rotation projection state of the molecule changes continuously in a controllable way. The effect of this type of magnetic manipulation on the rotation projection states is illustrated schematically in Fig. 1c using a simple classical picture.

After reaching the sample, a certain fraction of the molecular beam is scattered into the second arm of the instrument, by controlling the incident and final scattering angles through rotating the sample. In all the experiments presented below the sample was set to specular scattering conditions. Hereafter, the two types of measurements deviate. In one type of measurement, the second arm is used simply as a flux detector (see Supplementary Note 4 and Supplementary Fig. 7 for more details). In this case, the count rate at the particle detector is proportional to the flux of molecules that enter the second arm regardless of their quantum state, and it is only the first arm that is used to manipulate the quantum state of the impinging molecules. In the second scheme, which we call the full-interferometer scheme, the scattered beam goes through an additional electromagnet (B2), which is identical in design to the first electromagnet and allows us another opportunity to manipulate the components of the wave function, this time that of the scattered molecules. Finally, the beam goes through a second

magnetic lens, which selects the quantum states that are focused and enter the particle detector located at the end of the beam line.

Figure 2a,b presents the results of flux detection measurements for $H_2$ scattered from both a flat (111) and a densely stepped (115) copper surface. The flux measured by the particle detector as function of the magnetic field strength, B1, is shown in Fig. 2a after normalizing it to its average value. As we scan the magnetic field, we continuously change the rotation projection states of the molecules that reach the surface. The sample was oriented in such a way that only molecules that are scattered in the specular diffraction channel reach the detector. Thus, any intensity modulations of the measured flux reflect differences between the specular scattering of the different $m_J$ states, which we use as a probe of the stereodynamics of the surface-scattering process. The magenta line shows the scattered flux from a Cu(111) surface. While the signal is nearly flat, a weak oscillation pattern can be observed above the noise level at low field values, also seen as a small peak at $\sim 2.5\,\mathrm{kHz\,gauss^{-1}}$ in the corresponding spectrum (magenta trace in 2b).

A much clearer manifestation of steric effects is observed when $H_2$ molecules scatter from the highly corrugated stepped copper surface, Cu(115), shown as black and green curves in Fig. 2a,b. The Cu(115) surface is a stepped surface with narrow terraces separated by single atomic steps. When we modulate the quantum state of the impinging molecules, we observe large flux oscillations of the scattered beam. Furthermore, the strongly non-isotropic corrugation of the 115 surface leads to significant differences in the stereodynamics when the scattering plane is parallel or perpendicular to the direction of the atomic steps of this surface (black and green curves in Fig. 2a,b).

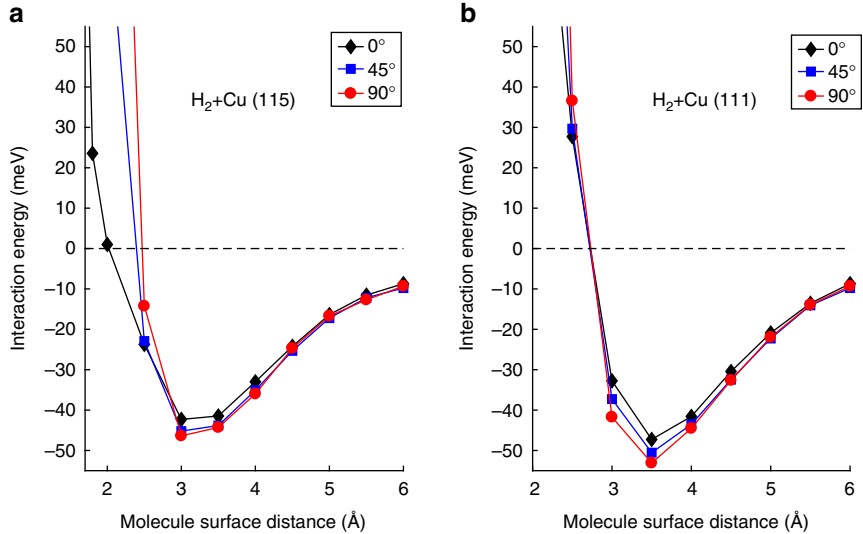

**Figure 3 | Density functional theory calculations.** (**a**,**b**) DFT calculations comparing the effect of the alignment of the $H_2$ molecule on the interaction with Cu(115) and Cu(111). The polar angles of orientation with respect to the surface normal are given in the legend (see also Supplementary Fig. 1).

**Computational**. To interpret our results we combined calculations of the propagation of the molecular wave functions through the magnetic fields of the apparatus with quantum dynamic calculations of the scattering event. Full details of these calculations can be found in Supplementary Notes 1–3; here we only provide a general description. The Ramsey Hamiltonian was used to calculate the evolution of the internal states (spin and rotation) within the field B1, and obtain the (superposition) quantum state of the molecule when it arrives at the sample surface. To include the stereodynamic scattering we chose a quantization axis, which is normal to the sample surface, and within that reference frame, we multiplied the modulus square of the arriving molecular amplitude, projected on each rotational substate (after summing over the three different $m_I$ spin substates), with its corresponding specular scattering probability to provide us with a simulated flux detection curve as function of B1. These curves were averaged over different velocities in the beam and also over different initial quantum states, where the weight of each initial quantum state was given by the transition probability through the first magnetic lens (Supplementary Table 2), calculated using a semi-classical deflection method[17]. It is important to note that, in the absence of a strong magnetic field, $m_J$ is not a good quantum number. Nevertheless, the projection onto the rotational substates, used to describe the scattering event, can be easily justified, given the negligible mixing of $m_I$ and $m_J$ states expected within the short duration of the scattering event, and the expectation that the nuclear spin state $m_I$ should not affect the scattering probability.

The differential scattering probabilities of the different rotation projection states (the projection axis being the surface normal) were obtained for $H_2$ scattering from Cu(111) by quantum dynamics calculations using a model[22] in which $H_2$ scatters from an ideal, static Cu(111) surface within the Born–Oppenheimer approximation. The potential energy surface was computed using density functional theory (DFT), employing the optPBE-vdW exchange correlation functional that gives an approximately correct description of the attractive van der Waals interaction[23] and accurately describes reactive scattering of $H_2$ from Cu(111) (ref. 24). Further details of these calculations are given in Supplementary Notes 1–2, Supplementary Figs 1 and 2 and Supplementary Table 1.

For the experimental value of the total incidence energy, and an incidence angle of 22.5 degrees, our calculations find that the specular scattering probabilities obtained for the $m_J = 0$ state on the one hand, and the $m_J = 1$ and $-1$ states on the other hand (the probabilities being the same for the latter two states) differ by $\sim 2.7\%$. For lower total incidence energies and, correspondingly, larger incidence angles, the ratio is even closer to 1 (see Supplementary Fig. 3). This is consistent with previous studies illustrating that at low total energies the $H_2$–Cu(111) interaction only exhibits a weak dependence on the orientation of $H_2$ relative to the surface[25], as also shown by our DFT calculations (Fig. 3b). As a result, the $m_J = 0$ and $m_J = \pm 1$ states of $H_2$ effectively see a very similar interaction potential and corrugation.

Figure 2c,d shows the simulated signal for $H_2$–Cu(111) and its corresponding spectrum using the calculated specular scattering probabilities mentioned above. The dominant peak in both the simulated and the experimental spectra is at 2.5 kHz gauss$^{-1}$, that is, the calculated expectation values for the rotation projections of the molecule oscillate at the same basic frequency as our measured scattered flux. However, when comparing the absolute modulation amplitudes we do notice a discrepancy between the calculated and measured results, indicating that the difference between the scattering probabilities of different rotational projection states are significantly underestimated by the calculation. This suggests that the corrugation of the molecule–surface interaction is more dependent on molecular orientation than theory predicts.

Quantum dynamical calculations for $H_2$ scattering from stepped metal surfaces are hard and therefore still rare, with only one quantum dynamics calculation performed so far for $H_2$–Pt(211) (ref. 26). However, the main difference we measured between Cu(111) and Cu(115) can be explained on qualitative grounds. As observed earlier for $H_2$–Cu(510) (ref. 27), $H_2$ may come in close proximity with the surface atoms positioned at the top of the steps on Cu(115). As a result, the interaction depends strongly on the orientation of the molecule relative to the surface at energetically accessible molecule–surface distances (Fig. 3a). In contrast, the anisotropy of the interaction with the terrace sites remains weak because at these sites $H_2$ cannot get close to the surface, as also observed for the flat (111) surface (Fig. 3b). On Cu(115) this leads to a strong corrugation of the molecule–

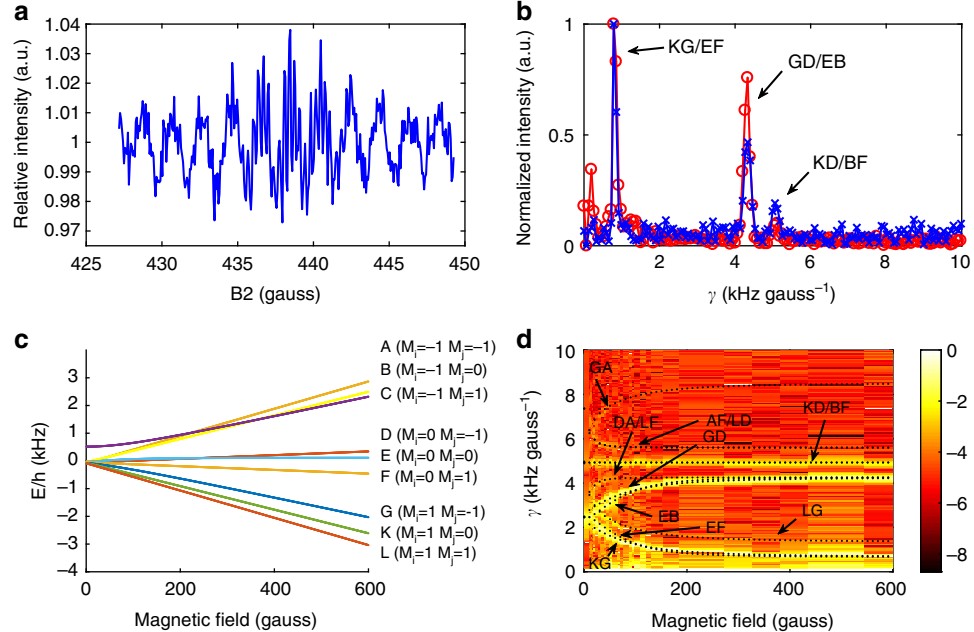

**Figure 4 | Full-interferometer mode.** (**a**) Measurements in full-interferometer mode close to the spin echo condition (B1 = 437 gauss). (**b**) Spectra of measurements close to the spin echo condition measured on Cu(111) and Cu(115) (blue crosses and red circle markers). (**c**) Field dependence of the Ramsey eigenenergies[20]. (**d**) 2D map created from concatenating spectra measured at different magnetic fields and plotting the logarithm of the intensity. The frequency peaks appear as high-intensity (yellow) bands. All the high-intensity frequency bands we measured fit the position of the Ramsey transitions superimposed on the image (black dots) and labelled using the scheme in **c**.

surface interaction, and, additionally, to a strong anisotropy of this corrugation. As a result, $m_J = 0$ and $m_J = |1|$ should effectively experience significantly different interaction potentials, providing a possible explanation for the much stronger oscillations observed in the specular scattering flux for the Cu(115) surface as the initial $m_J$ state is modulated.

Flux detection measurements of the type we present in Fig. 2 demonstrate that it is now possible to measure and compare the role of steric effects in various molecule–surface systems through their effect on the differential scattering probabilities. One obvious extension of our study is to perform further measurements on different molecule–surface systems and at different conditions (molecular velocity, surface temperature, sample orientation and so on), in order to supply the much-needed experimental benchmarks for developing theoretical models of molecule–surface interactions. Below, we briefly present another direction in which this technique can be extended, which involves using the set-up in full-interferometer mode.

**Full-interferometer measurements**. To perform full-interferometer measurements we fix one field (B1 or B2) at a constant value and scan the other field within a small range around the centre value of B2 = −B1. This type of measurement is the molecular analogue of a spin–echo scan in neutron or atom interference experiments, where the reverse action of the second field (B2 ≈ −B1) leads to constructive interference that is unaffected (to first order) by the velocity spread of the particles in the beam[15,28]. Figure 4a presents the results of such measurements for H$_2$ scattering from Cu(111). There is a rich oscillation pattern in the molecular interference measurement, due to the Rabi oscillations between different Ramsey states, which take place within the magnetic field we scan. The blue crosses in Fig. 4b show a Fourier transform of this signal. The frequency axis was converted to kHz gauss$^{-1}$ (labelled $\Upsilon$), which has the units of

a gyromagnetic ratio. The spectrum has three important dominant peaks at 0.76, 4.28 and 5 kHz gauss$^{-1}$ (the increase in intensity closer to the origin is at least partially due to low-frequency drifts in the detector background signal).

The positions of the $\Upsilon$ peaks change when we perform the spin echo measurements at different magnetic fields. This can be understood from looking at the Ramsey eigen energies plotted in Fig. 4c, illustrating that unlike the simple case of spin ½ particles, the field dependence of these energies is not a simple linear relation. Figure 4d is a two-dimensional (2D) colour plot that combines spectra of the type shown in Fig. 4b (that is, scanning B2 around −B1) for B1 values ranging from 0 to 600 gauss. The peaks of individual spectra combine into multiple high-intensity bands, which are characterized by different magnetic field-dependent frequencies and different amplitudes. The frequencies of all the interference oscillations we measured can be assigned to transition frequencies, $f_{i,j}(B) = |\frac{d\Delta E_{i,j}(B)}{dB}\frac{1}{h}|$, where $\Delta E_{i,j} = E_i - E_j$ is the energy difference between each pair of Ramsey states (see Supplementary Note 3 and Supplementary Fig. 5). The 12 transitions that follow the nine high-intensity bands seen in our measurements are marked with black dots in Fig. 4d and labelled using Ramsey's notation scheme (shown in Fig. 4c). Thus, each peak in the one-dimensional (1D) or 2D spectrum can be assigned to a pair of Ramsey states, and allows us to follow the evolution of these states both before and after scattering.

Of the 36 possible transition frequencies, only 12 (at most) have non-vanishing amplitudes and within this subgroup the amplitudes vary considerably (Fig. 4d shows the logarithm of the intensity). The relative amplitude differences, and in particular the vanishing amplitudes of some of the transition frequencies hold key information regarding the state-to-state scattering event. This point is demonstrated experimentally in Fig. 4b when comparing the full-interferometer mode spectra measured on Cu(111) (blue crosses) with the result obtained from an identical experiment on a Cu(115) surface (red circles). The relative

amplitudes are substantially different. Since in both measurements we scanned the second field, B2, the dissimilar interference patterns reflect the differences in the scattered quantum phase because of differences in the stereodynamic interaction.

The relative intensities of the peaks in the full-interferometer-type measurement are related to both changes in the quantum state population (for example, rotational excitation) and destructive and constructive interference of the different wave packets due to the phase acquired in the scattering process. In Supplementary Note 3 and Supplementary Fig. 6 we present calculations that demonstrate the sensitivity of the relative amplitudes to the phase changes in the scattered quantum state. Interpretation of the experimental interferograms of the type shown in Fig. 4b,d in terms of relative amplitudes is therefore a challenging theoretical task. Extracting the details of the interaction potential will require the application of efficient techniques for solving the inverse scattering problem in a multidimensional parameter space (for example, ref. 29). Nevertheless, full-interference mode measurements contain very valuable information about the changes to the quantum state of the molecule due to the scattering process, which we believe warrant future efforts directed at developing the required interpretation capabilities.

## Discussion

In summary, we have presented a general experimental approach for controlling, and studying the role of, the rotation projection states of a molecule scattering from a surface. We applied the approach to study the scattering of $H_2$ from flat and highly corrugated copper surfaces. Using the flux detection mode of the apparatus we demonstrated that we can change the specular scattering intensity by manipulating the rotation projection states of the impinging molecular beam. More generally, our approach allows us to study the dependence of any scattering probability on the rotation projection states. The trend we observe, that the steric effect is significantly more dominant on stepped surfaces, is consistent with theoretical considerations. The absolute differences in the probabilities produce a challenging benchmark for further improvement of the calculated molecule–surface interaction and theoretical models. Finally, we also presented full-interferometer mode measurements, an extension of the experimental technique that is sensitive to the quantum states of the molecule after the scattering event. While we present a first application to scattering of $H_2$, the technique only relies on the rotation of the molecules in a magnetic field and should work well for many molecules. Furthermore, in our experiments we controlled the outcome of a scattering event by modifying B1. While this still needs to be demonstrated, it seems likely that the reaction probability of molecules can be manipulated in a similar way, if the reactions depend on the rotational alignment or orientation of the impinging molecules, opening new opportunities for controlling reaction yields.

## Methods

**Molecular beam source.** The continuous molecular beam is produced by a supersonic expansion through a 20-micron nozzle. Using a $H_2$ gas pressure of 5 bars and a nozzle temperature of 100 K, we obtained a beam with a mean velocity of 1450 m s$^{-1}$ and a full width half maximum of 4%. The velocity distribution was derived from the position and profile of the diffraction peaks obtained when scattering from the Ni(111) surface (the diffraction intensities were too weak to measure on the Cu(111) surface).

**Magnetic lenses.** The first hexapole field/magnetic lens consists of 6 Halbach-type hexapole field magnets, following the design of Jardine *et al.*[17]. This design is optimized to maintain adequate pumping of the defocused beam particles, which is crucial for the high gas load conditions close to the beam source. Adjacent to the last hexapole element we installed a hexapole–dipole field transmission element,

which adiabatically rotates the magnetization towards a particular axis in the lab frame (the dipole field) and defines the quantization axis of the magnetic lens element, similarly to the scheme used in the HeSE experiment[30].

The second hexapole field/magnetic lens consists of 16 Halbach-type hexapole field magnets, following the design of Dworski *et al.*[31], optimized for strong deflection and small magnetic aberrations. The dipole–hexapole transmission element in this case is attached to the first magnet of the hexapole assembly, defining the quantization axis of the magnetic state selection process before the molecules enter the hexapole field region where the trajectories split in space.

**Electromagnetic fields.** The two electromagnets producing the B1 and B2 fields are 1-m-long solenoids producing a total field integral of 0.011 Tesla × metre at a current of 1 Amp. The winding pattern was optimized to produce strong field integrals while maintaining a high field integral homogeneity for different beam trajectories ($<1$ p.p.m.). The electromagnets are shielded by a triple layer of Mu metal, where the inner layer serves as a flux return layer and the two outer layers shield from external magnetic field perturbations. Heat dissipation is provided using water cooling flow, underneath and above the solenoid windings. High-stability control of the two fields is achieved by two independent high-stability power supplies (Danfysik) calibrated to control currents on a p.p.m. level over a 0–10A range.

**Sample preparation and sample holder.** The single crystal copper samples (Surface preparation laboratory, the Netherlands) were mounted on a non-magnetic six axis sample manipulator with cooling, heating and sample transfer capabilities, inside a ultra-high vacuum magnetically shielded chamber (base pressure $<10^{-10}$ mbar). The samples were cleaned by cycles of sputtering (1,000 eV Ar +, 10 min × 10 μA at 300 K and annealing at 800 K). The sample temperature was stabilized at $200 \pm 1$ K using a combination of liquid $N_2$ cooling and radiative heating.

**Particle detector.** An ultra-high efficiency particle detector was used at the end of the beam line. The detector ionizes the hydrogen molecules using electron bombardment in a magnetic trap, followed by magnetic-sector separation of mass to charge ratio, and finally an electron multiplier that measures the current. The detector is an improved version of a previous magnetic trap detector[32] and was designed by the SMF group at the Cavendish Laboratory, University of Cambridge.

**Data availability.** The data that support the findings of this study are available from the corresponding author upon request.

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

## Acknowledgements

We thank Dick Manson, Bill Allison, Daniel Farias and Aart Kleyn for insightful discussions. We are grateful to Mark Wijzenbroek for providing the potential energy surface for $H_2 + Cu(111)$. This work was funded by the German-Israeli Foundation for Scientific Research, the Natural Sciences and Engineering Research Council of Canada, the Netherlands Organisation for Scientific Research (NWO) through Vidi grant 723.014.009, the European Research Council under the European Union's seventh framework programme (FP/2007-2013)/ERC grant 307267 and by the European Research Council through an ERC-2013 advanced grant 338580.

## Author contributions

G.A. conceived and designed the experiments. O.G., G.C. and Y.A. converted the HSE set-up for $H_2$ and performed the measurements and data post processing. T.M. developed the time-dependent model. T.M./O.G. performed the analytic/numerical calculations using this model. J.T.C. developed and performed the transfer matrix analysis and the corresponding numerical calculations. R.V.K. contributed to the development of the transfer matrix analysis. M.F.S. performed the quantum dynamics calculations of $H_2$/Cu(111). J.M. performed the DFT calculations on $H_2$/Cu(511). G.-J.K. supervised the research of M.F.S. and J.M. The manuscript was written by G.A., G.-J.K. and R.V.K. The SI contributions were written by G.A., M.F.S., J.M., J.T.C., R.V.K., T.M. and G.A. All the authors made significant contributions to the interpretation of the results and commented on the manuscript.
