## [Peer Review File · Nature Communications]

Reviewers' comments:

Reviewer #1 (Remarks to the Author):

This paper describes an experimental-theoretical collaboration that establishes a new approach to unraveling the stereo dynamics of molecule-surface scattering. Detailed computational work supports and provides insight into the experimental method. The results are novel and will be of significant interest to the surface dynamics community. The ability to measure and describe this fundamental process has broader implications for our ability to understand, predict, and potentially control the gas-surface interactions.

The experiments provide both total flux measurements, which monitor the angle-resolved scattered flux of molecules as a function of the rotational projection states of the incident beam, and interferometric measurements that reveal the quantum state identity of the scattered states as well. The interferometric measurements also yield insight into phase shifts of the wave function that occur upon scattering.

Theory models the state selection scheme that defines the incident flux of rotational projection states, and then, using previously established quantum dynamics methods, makes predictions about the scattered molecule flux. The agreement between experiment and theory is solid for the total flux measurements. Calculations simulating scattering for the full interferometric measurements are complicated by the high sensitivity of scattering amplitudes to subtle phase shifts to the wave function that occur upon scattering. Taken together, the work highlights the high level of detail available in the experimental measurements, and makes a strong argument that details in the scattering amplitudes can be used to further refine theoretical models of the gas-surface scattering potential.

Computational modeling of the H₂/Cu system has been a benchmark for quantum dynamics calculations for many years, and these experiments provide another platform for benchmarking the quality of calculated potentials and of the quantum dynamics methods used to simulate scattering. The potential ability to extend this work to other molecule surface systems will help provide benchmark data essential for extending computational methods to more chemically complex systems.

There is currently interest in whether the magnetic properties of some metals (e.g. Ni) impact their catalytic activity. For the future, the authors may consider whether insights and methods developed for this work could be used to address that point.

Many of the experimental methods and design have been discussed in prior (cited) publications. Details specific to this work were described in the manuscript or in the supplemental section. Extensive details on computational methods and models were clearly laid out in the lengthy supplemental section as well as in cited work.

Typos, minor comments -

on P. 8, 1st paragraph - Figure comparing scattering probabilities for $m_j=0$ and 1 states appears to be Fig. S3, not S4.

In summary, I found the work to be original and important, and I recommend publication without the need for revision.

Reviewer #2 (Remarks to the Author):

I think that the authors present an important advance in what might be called stereochemistry. However, I am unable to recommend publication of the present manuscript until some confusion

on my part is cleared up. I think Table S2 is quite important, if I understand it, and tells me that all nine possible m subscript i and m subscript j states are transmitted to the surface but with different probabilities. Is this correct?

If not, please make clear what is transmitted to the surface for subsequent scattering analysis.

If true, then why does the manuscript seem to discuss only $m_j = 0$ and $m_j = 1$ and not $m_j = -1$?

In a strong magnetic field the nuclear spin and rotational angular momenta are decoupled but in the absence of a magnetic field, they recouple and the good quantum numbers are F and m_F . This is not made clear in the manuscript, at least, not to me. What is the proper way I should think about what strikes the surface?

Reviewer #3 (Remarks to the Author):

In this article, Godsi et al present a detailed and compelling study of molecular scattering of H_2 from Cu surfaces. While many such experiments have been reported in the past, the presented work is unique in that it distinguishes the rotational-magnetic quantum state of the incoming and, in some cases, the scattered molecular beam, and therefore measures specific scattering cross sections between these states. While oriented molecular scattering experiments have been carried out before, they have relied on special cases where the molecule contained paramagnetic entities, or a permanent electric dipole. The approach described by Godsi et al is novel to the best of my knowledge. It uses the coupled rotational-nuclear magnetic quantum states of the scattered molecule, which evolve in a predictable way during propagation through a homogeneous magnetic field under the Ramsey Hamiltonian. Specific nuclear spin-rotation states can be selected by a Stern-Gerlach magnet.

The manuscript is very carefully and clearly written, and a joy to read.

The only possible weak point in the paper is the poor agreement between the predicted and observed modulation of the scattering amplitude as a function of B_1 field. A more in-depth discussion of the possible failure points of the theoretical model employed would have been interesting, but, in the interest of compactness of the paper, is probably best deferred to a separate publication. At any rate, the novelty and elegance of the experimental approach and the convincing experimental results easily justify publication in Nature Comm. and I recommend publication without change.

Below, referee comments are in blue, our responses in black and added/modified text in red.

Reviewer #1 :

This paper describes an experimental-theoretical collaboration that establishes a new approach to unraveling the stereo dynamics of molecule-surface scattering. Detailed computational work supports and provides insight into the experimental method. The results are novel and will be of significant interest to the surface dynamics community. The ability to measure and describe this fundamental process has broader implications for our ability to understand, predict, and potentially control the gas-surface interactions.

The experiments provide both total flux measurements, which monitor the angle-resolved scattered flux of molecules as a function of the rotational projection states of the incident beam, and interferometric measurements that reveal the quantum state identity of the scattered states as well. The interferometric measurements also yield insight into phase shifts of the wave function that occur upon scattering.

Theory models the state selection scheme that defines the incident flux of rotational projection states, and then, using previously established quantum dynamics methods, makes predictions about the scattered molecule flux. The agreement between experiment and theory is solid for the total flux measurements. Calculations simulating scattering for the full interferometric measurements are complicated by the high sensitivity of scattering amplitudes to subtle phase shifts to the wave function that occur upon scattering. Taken together, the work highlights the high level of detail available in the experimental measurements, and makes a strong argument that details in the scattering amplitudes can be used to further refine theoretical models of the gas-surface scattering

potential.

Computational modeling of the H₂/Cu system has been a benchmark for quantum dynamics calculations for many years, and these experiments provide another platform for benchmarking the quality of calculated potentials and of the quantum dynamics methods used to simulate scattering. The potential ability to extend this work to other molecule surface systems will help provide benchmark data essential for extending computational methods to more chemically complex systems.

There is currently interest in whether the magnetic properties of some metals (e.g. Ni) impact their catalytic activity. For the future, the authors may consider whether insights and methods developed for this work could be used to address that point.

This is a good idea and will definitely be pursued in future studies.

Many of the experimental methods and design have been discussed in prior (cited) publications. Details specific to this work were described in the manuscript or in the supplemental section. Extensive details on computational methods and models were clearly laid out in the lengthy supplemental section as well as in cited work.

Typos, minor comments -

on P. 8, 1st paragraph - Figure comparing scattering probabilities for $m_j=0$ and 1 states appears to be Fig. S3, not S4.

The referee is right, the typo was corrected on page 8.

In summary, I found the work to be original and important, and I recommend publication without the need for revision.

We thank the referee for his positive comments.

Reviewer #2 (Remarks to the Author) (**note that we have numbered the comments of the referee below**):

1. I think that the authors present an important advance in what might be called stereochemistry. However, I am unable to recommend publication of the present manuscript until some confusion on my part is cleared up. I think Table S2 is quite important, if I understand it, and tells me that all nine possible m subscript i and m

subscript j states are transmitted to the surface but with different probabilities. Is this correct?

Not quite. Indeed the all nine (m_l, m_j) states are transmitted by the first lens towards the sample, but with different probabilities given in table S2. Also, their amplitudes and those of the other states are further changed in the first arm of the apparatus: please see below under the comment labeled "5".

2. If not, please make clear what is transmitted to the surface for subsequent scattering analysis.

Please see below under the comment labeled "5".

3. If true, then why does the manuscript seem to discuss only $m_j=0$ and $m_j=1$ and not $m_j=-1$?

Obviously both $m_j=1$ and $m_j=-1$ molecules reach the surface, and the reason only $m_j=1$ and $m_j=0$ were plotted in Fig. S3 was that the scattering probabilities of $m_j=1$ and $m_j=-1$ are equal by symmetry. We agree that this point was not clear in the original manuscript and have added a sentence on page 8 saying this explicitly, as well as using the relation $m_j=|l|$ instead of $m_j=1$ on page 9. Changes have also been made to Section 2 of the Supporting Information to make this point clearer.

4. In a strong magnetic field the nuclear spin and rotational angular momenta are decoupled but in the absence of a magnetic field, they recouple and the good quantum numbers are F and m_F . This is not made clear in the manuscript, at least, not to me.

Indeed in the absence of a strong magnetic field the nuclear spin and the rotational magnetic moment couple. While our calculation takes this coupling into account, we agree that this was not made clear enough in the manuscript and we have altered the text at the end of page 4 and beginning of page 5 to emphasize this and added a reference to an experimental study (Phys. Chem. Chem. Phys. 11, 142–147 (2008)) which studied the mixing rate in HD, and nicely demonstrates this.

The new text starts at the bottom of page 4 and continues onto page 5, and reads:

“...However it is important to note that generally speaking, a magnetic focusing element cannot be used on its own as a method of selecting a pure rotational projection state, due to the spin-rotation coupling terms [20] that lead to mixing of the rotational states even in the absence of magnetic fields [21]. Instead the lens selects the initial quantum state of the molecule, which then evolves into a superposition of m_I and m_J states as the molecule travels through the beam line, as discussed in more detail in the supplementary information section.

The second crucial element is an electromagnet (B1) which is used to control the evolution of the quantum (superposition) state mentioned above, according to the Hamiltonian... ”

The referee may also look at the revised Section 3 of the SI, where all this is explained in detail.

5. What is the proper way I should think about what strikes the surface?

While strictly speaking a super position of m_I , m_J states is needed to describe the wave function of the molecule hitting the surface due to the absence of a strong magnetic field (see response to last comment), the wave function can be projected onto the rotational sub-states for calculating the scattering event due to the fact that the scattering event is many orders of magnitude faster than the mixing time between m_I and m_J states. We agree that this point was not clear in the original manuscript and have added the following paragraph accordingly.

We inserted the following new text on page 7:

” It is important to note that in the absence of a strong magnetic field, m_J is not a good quantum number. Nevertheless , the projection onto the rotational sub-states used to describe the scattering event, can be easily justified given the negligible mixing of m_I and m_J states expected within the short duration of the scattering event, and the expectation that the nuclear spin state m_I should not affect the scattering probability.”

Reviewer #3 (Remarks to the Author):

In this article, Godsi et al present a detailed and compelling study of molecular scattering of H2 from Cu surfaces. While many such experiments have been reported in the past,

the presented work is unique in that it distinguishes the rotational-magnetic quantum state of the incoming and, in some case, the scattered molecular beam, and therefore measures specific scattering cross sections between these states. While oriented molecular scattering experiments have been carried out before, they have relied on special cases where the molecule contained paramagnetic entities, or a permanent electric dipole. The approach described by Godsi et al is novel to best of my knowledge. It uses the coupled rotational-nuclear magnetic quantum states of the scattered molecule, which evolve in a predictable way during propagation through a homogeneous magnetic field under the Ramsey Hamiltonian. Specific nuclear spin-rotation states can be selected by a Stern-Gerlach magnet.

The manuscript is very carefully and clearly written, and a joy to read. The only possible weak point in the paper is the poor agreement between the predicted and observed modulation of the scattering amplitude as a function of B1 field. A more in-depth discussion of the possible failure points of the theoretical model employed would have been interesting, but, in the interest of compactness of the paper, is probably best deferred to a separate publication. At any rate, the novelty and elegance of the experimental approach and the convincing experimental results easily justify publication in Nature Comm. and I recommend publication without change.

We thank the referee for his positive recommendation. We agree with the referee that a further in-depth study of the deviation between the theoretical predictions and measurements of the steric dependency of the scattering amplitudes are very interesting, and we are actively working on developing a theory that can predict the outcome of this scattering experiment. However, we also agree with the referee that such studies are best deferred to a separate publication.

REVIEWERS' COMMENTS:

Reviewer #2 (Remarks to the Author):

Thanks for the clear explanations about the concerns I had, which are fully satisfied. I am pleased to recommend publication without any further delay.